# SEAL: Scaling to Emphasize Attention for Long-Context Retrieval

## Abstract

In this work, we introduce a novel approach called Scaling to Emphasize Attention for Long-context retrieval (SEAL), which enhances the retrieval performance of large language models (LLMs) over extended contexts. Previous studies have shown that each attention head in LLMs has a unique functionality and collectively contributes to the overall behavior of the model. Similarly, we observe that specific heads are closely tied to long-context retrieval, showing positive or negative correlation with retrieval scores. Built on this insight, we propose a learning-based mechanism using zero-shot generated data to emphasize these heads, improving the model's performance in long-context retrieval tasks. By applying SEAL, we can achieve significant improvements in in-domain retrieval performance, including document QA tasks from LongBench, and considerable improvements in out-of-domain cases. Additionally, when combined with existing training-free context extension techniques, SEAL extends the context limits of LLMs while maintaining highly reliable outputs, opening new avenues for research in this field.

## 1 Introduction

Large Language Models (LLMs) (Brown et al. (2020), Radford et al. (2019), Touvron et al. (2023)) are capable of rapidly generating high-quality answers to a wide range of questions by leveraging the diverse knowledge embedded in their vast number of parameters. However, in-depth analyses have revealed a common issue known as hallucination (Shuster et al. (2021), Lin et al. (2021), Ji et al. (2023)), where the models confidently produce inaccurate answers. To address this, research has focused on using external information as context to guide the outputs, such as Retrieval-Augmented Generation (Lewis et al. (2020), Xu et al. (2023)) and Chain-of-Thought reasoning (Wei et al. (2022)). These approaches have significantly improved the reliability of LLMs by enabling them to reference existing information during generation. However, this trend has also highlighted a key limitation of LLMs: the constraint of their context window length.

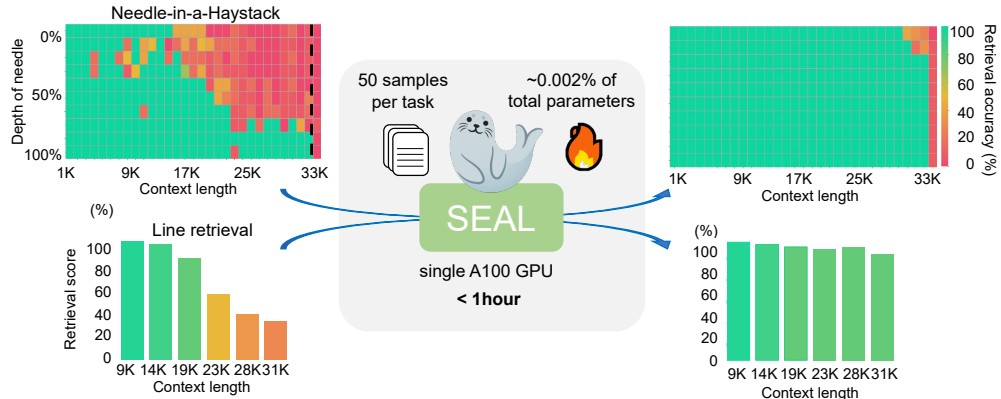

Figure 1: Overview of the proposed SEAL and corresponding retrieval score improvements for LongChat-7B-v1.5-32K (Li et al. (2023)) model.

This limitation of the context window stems from several problems, such as the design constraints of positional encoding (Su et al. (2024)) in LLMs and the preference for shorter sequences in training data. It is an inherent feature of trained LLMs, where performance rapidly degrades once the predefined context window size is exceeded. To mitigate this issue, several training-free and fine-tuning-based methods (Xiao et al. (2023), Han et al. (2023), Zhang et al. (2024)) have been developed to extend the context length of trained LLMs. Recently, model providers have even started releasing models specifically designed for long-context to address this limitation (Abdin et al. (2024), Jiang et al. (2024)).

However, even within extended context windows, performance tends to degrade as the context length approaches its limit. This leads to phenomena such as the "lost in the middle" (Liu et al. (2024a)), where the model exhibits biases toward focusing on the early and later parts of the context, resulting in an increased likelihood of incorrect answers or hallucinations. Including this phenomenon, issues in which retrieval performance is influenced by input length have been consistently observed.

In this study, we aim to address this second problem. We specifically address cases where retrieval tasks are performed on long-context inputs, which we define as long-context retrieval. Our approach is based on the insight that well-trained LLMs possess the inherent ability to infer information accurately regardless of context length, but biases in their trained parameters often lead to performance degradation. For a representative long-context retrieval benchmark, we observed that certain attention heads contribute notably to long-context retrieval, adjusting their strength to either improved or reduced accuracy largely.

Built on these observations, we propose a novel approach, *Scaling to Emphasize Attention for Long-context retrieval (SEAL)*. SEAL is a learning-based attention scaling technique that fine-tunes attention strength using stochastic gradient descent (SGD) on a small set of generated data following the format of the task domain. SEAL consists of two major processes. First, training data focused on the context format are generated for the target task. Our goal is to alter the head-wise contribution rather than update the embedded knowledge. Therefore, a small set of generated data is sufficient to identify the important heads relevant to retrieval. Subsequently, head-wise and channel-wise learnable scales are fine-tuned for SEAL-H (head) and SEAL-C (channel), respectively. Through this process, SEAL not only probes the importance of each attention component but also adjusts the scaling to enhance retrieval performance. Unlike widely known Parameter-Efficient Fine-Tuning methods (Hu et al., Houlsby et al. (2019)), SEAL focuses on emphasizing the heads relevant to retrieval, supported by our observations, which enables high accuracy with minimal data and learnable parameters.

Using SEAL, we have achieved significant accuracy improvements in in-domain environments with less than one hour of fine-tuning for 7B models, regardless of the model type. Additionally, we have verified that SEAL maintains generalization ability even for out-of-domain tasks. Most importantly, SEAL has delivered substantial improvements in long-context retrieval accuracy for LLMs that had already been trained and had their context extended using existing techniques, and this breakthrough opens up new possibilities for enhancing the long-context retrieval capabilities of existing LLMs.

## 2 RELATED WORK

**Circuit Analysis**   There have been continuous efforts to identify and interpret the internal mechanisms of LLMs and Transformers. Elhage et al. (2021) analyzed the mechanism of a two-layer attention-only model, revealing the presence of attention heads that contribute to in-context learning. Ferrando et al. (2024) identified various roles of attention heads, such as copy heads and positional heads. Wu et al. (2024) further demonstrated that certain heads play a role in copying the correct answer during retrieval. These studies have primarily focused on analyzing the roles of individual heads, and in addition, analysis methods such as circuit analysis and logit attribution (Ferrando et al. (2024), Lieberum et al. (2023)) have been proposed.

**Context Window Extension**   There are several studies to push beyond the limitations of LLMs' pre-trained context window. Position interpolation-based methods (Chen et al. (2023), Peng et al. (2023)) have been proposed for models using Rotary Position Embedding (RoPE) (Su et al. (2024)), where interpolation is applied to position encodings and then fine-tuned with a small amount of data. Alternative methods have been proposed to increase the context length based on the Neural Tangent Kernel (bloc97 (2023a), bloc97 (2023b), emozilla (2023)) theory, which takes into account the loss

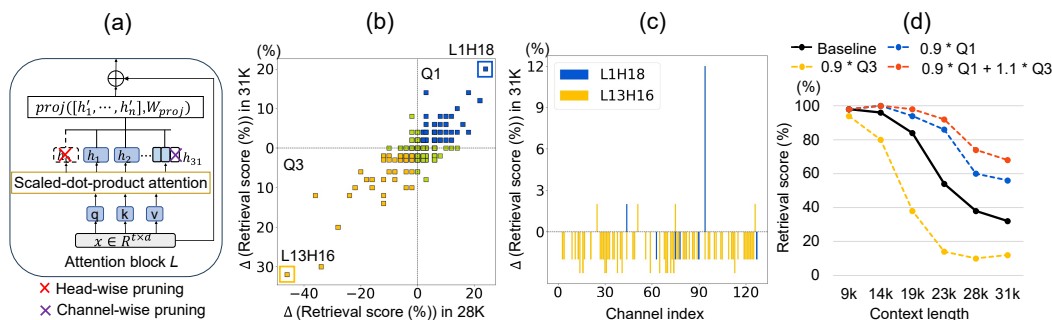

Figure 2: Changes in retrieval scores (%) with different settings. (a) Overview of pruning settings, (b) head-wise pruning results, (c) channel-wise pruning results, and (d) retrieval scores of scaling multiple heads. L$x$H$y$ refers to the $y$-th head of the $x$-th Transformer block (zero-based indexing).

of information at high frequencies. Self-Extend (Jin et al. (2024)) introduces grouped positions to map positions beyond the learned context length to positions within the learned context, allowing it to handle long input without additional training.

**Benchmarks for Long-Context LLMs** Several benchmarks have been proposed to evaluate the retrieval and reasoning capabilities of long-context LLMs. Needle-in-a-Haystack (Kamradt (2023)) inserts a random fact or statement ('needle') into a long-context text ('haystack') and asks the model to retrieve the needle. This benchmark has shown that LLMs struggle to retrieve the needle as the input context length increases. LongEval (Li et al. (2023)) line retrieval is the task of retrieving the corresponding digit given a key within a long text consisting of sentences with a line key and a value of up to five digits. LongBench (Bai et al. (2023)) is a benchmark consisting of 21 tasks across 6 categories, designed to comprehensively assess long-context understanding capabilities.

## 3 MOTIVATION

Research on Transformer-based architectures (Elhage et al. (2021), Ferrando et al. (2024)) has shown that attention heads, a key component, perform distinct roles such as copying, retrieval, and relevance, working together to shape the network's overall functionality. Notably, some heads specialize in handling long sequences, while others focus on retrieval. This leads to an optimistic prediction: **if we can identify and strengthen the heads specialized in long-context retrieval, we might significantly enhance performance in that area.**

### 3.1 PRIMARY OBSERVATION: PER-HEAD PRUNING

To validate this prediction, we first re-examined whether each attention head contributes differently to the retrieval process and determined if we could identify an attention head specialized for retrieval. Our experimental design is straightforward. As shown in Figure 2(a), we pruned one head at a time on the LongChat-7B-32K (Li et al. (2023)) model and compared the resulting accuracy changes with the accuracy of the baseline network. To simplify the experiment, we used the LongEval (Li et al. (2023)) line retrieval benchmark, where the goal is to retrieve a digit of up to five characters randomly located in a given text. This benchmark was particularly convenient because the target retrieval tokens are limited to the digits 0 through 9.

As shown in Figure 2(b), the impact of each head varied significantly, with accuracy changes of approximately ±20% or more, indicating that certain attention heads play a crucial role in retrieval. These positive and negative head-wise impacts were consistently observed in both mid-length (x-axis) and long-length (y-axis) contexts. While these results do not definitively show whether the heads are directly involved in retrieval or are performing other important tasks necessary for accuracy (*e.g.*, understanding the format), an intriguing observation emerges: pruning certain attention heads can actually lead to an increase in retrieval scores.

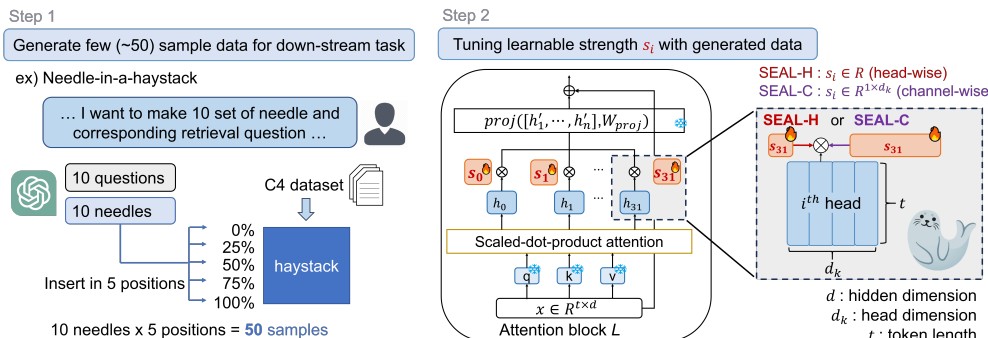

Figure 3: The overview of the proposed SEAL method. SEAL-H (head) or SEAL-C (channel) can be used depending on scaling granularity.

### 3.2 GENERALIZED APPROACH: ATTENTION HEAD-WISE SCALING

Next, we developed a more general approach to extend the head-wise pruning experiment. Since pruning multiple heads simultaneously can lead to performance degradation, a more scalable method was needed. To address this, we adjusted the scale of the identified heads to see if this could holistically improve accuracy. Built on this insight, we divided the quadrants in Figure 2(b) based on baseline performance (0.0) on the x and y axes. Instead of pruning individual heads, we tried scaling multiple heads together. By scaling the influence of all heads in the first quadrant (Q1)—whose pruning benefits the retrieval task—by 0.9, we observed an accuracy increase from 32% to 56% at the input length of 31K (blue dotted line in Figure 2(d)). In contrast, scaling the heads in Q3—whose pruning degrades retrieval—by 0.9 resulted in a significant drop in performance (yellow line). Interestingly, when we scaled Q1 by 0.9 and Q3 by 1.1 simultaneously, we observed an even greater improvement in retrieval scores (red line). This suggests that jointly scaling and controlling the influence of these heads can significantly enhance retrieval performance.

### 3.3 EXTENDED APPROACH: ATTENTION CHANNEL-WISE SCALING

While previous observations show that head-wise scaling offers new possibilities for improving long-context retrieval performance, there is still room for refining the granularity of scaling. As noted in Quantizable Transformers (Bondarenko et al. (2023)), earlier research suggests that specific channels handle syntactic elements like delimiter tokens, and even encode task-specific knowledge (Rudman et al.). In our LongChat-7B (Li et al. (2023)) pruning experiment, we further applied channel-wise pruning to the head with the greatest performance improvement (L1H18) and the head with the largest performance drop (L13H16), as shown in Figure 2(c). Interestingly, within L1H18's 128 channels, only certain channels accounted for most of the performance changes. Similarly, when we controlled L13H16 at a finer channel level, we discovered that some channels actually improved performance during pruning, though the overall head caused a significant drop. This underscores the need for channel-wise manipulation at a finer granularity than the head-level adjustments.

## 4 PROPOSED METHOD: SEAL

Built on these invaluable observations, we introduce a novel method called Scaling to Emphasize Attention for Long-Context Retrieval (SEAL), a framework designed to validate our findings and enhance the long-context retrieval performance of existing LLMs. In SEAL, we update existing LLMs without altering their learned behavior, instead efficiently adjusting the strength of each attention component. Since sequentially performing head or channel-wise pruning to identify the influence of all heads or channels for each task is infeasible, our key idea is to leverage gradient descent to ascertain the impact of each head on retrieval. Figure 3 provides an overview of SEAL. SEAL is intentionally designed to validate our observations and enables the updating of LLMs with minimal training data and fine-tuning, as outlined in the previous section. SEAL's key contributions are in

two main areas: context-aware generation of training datasets and the design of a learnable space that maximizes retrieval performance while minimizing cost.

### 4.1 GENERATING TRAINING DATA FOCUSED ON THE CONTEXT FORMAT

During the dataset generation stage, we observed that SEAL's focus is not on the inherent value of real-world data, but rather on the format of data representation for long-context tasks. To demonstrate this, we generated synthetic training data using an LLM and the task domain's format, instead of using real data with meaningful values, and used it to train the attention strength.

Initially, we generated 50 sample input and answer sets for the given downstream long-context task. To avoid contamination, we ensured consistency only in format while generating random content. The method for obtaining format samples may vary depending on the type of downstream task. The left side of Figure 3 visualizes the pipeline for generating training samples for the Needle-in-a-Haystack task, as an example. Below are examples created for line retrieval (a) and Needle-in-a-Haystack (b) tasks.

---

(a) **Prompt:** ... line righteous-ethernet: REGISTER_CONTENT is <40779> ...
**Answer_string:** The <REGISTER_CONTENT> in line righteous-ethernet is 40779.

(b) **Prompt:** ... Based on the content of the book, Question: What is immediately noticeable upon entering the room?
**Answer_string:** Immediately noticeable upon entering the room is the large oak table positioned beneath the chandelier.

---

### 4.2 LEARNABLE SPACE DESIGN: SEAL-H AND SEAL-C

Using the generated data, we trained a learnable scaling for attention components. Based on the intuition from pruning experiments of Section 3, we propose two granularities for attention control. The first is SEAL-H (head), which places a learnable scalar head-wise to learn the strength of each head (Figure 3 Right). This process allows us to probe the influence of each head on retrieval while jointly learning scaling appropriate for long contexts. The second option is SEAL-C (channel), which additionally uses a learnable vector for the hidden dimension of each attention output (channel-wise). As observed in Section 3.3, we found that within the attention heads, there are channels that have both positive and negative impacts. SEAL-C assigns and updates parameters on a per-channel basis. While this increases the number of parameters to be learned, it is expected to allow for more fine-grained manipulation of the attention head outputs, potentially leading to improved performance.

### 4.3 PEFT BASELINE: SEAL-L (LoRA)

The proposed SEAL method can be categorized under Parameter-Efficient Fine-Tuning (PEFT), as it selects vital, minimal learnable parameters that can impact retrieval and performs supervised fine-tuning on these scales. From this perspective, a representative PEFT, LoRA (Hu et al.), can intuitively serve as our baseline and validate the effectiveness of our fine-tuning pipeline. Furthermore, comparisons with SEAL-C and SEAL-H suggest that if these methods achieve performance comparable to SEAL-L with fewer parameters, it validates that we accurately identify the key factors contributing to improved retrieval performance. Considering the most basic form of LoRA with rank 1 ($r = 1$), the learnable vectors of LoRA adjust the retrieval-related influence in a manner similar to SEAL-C by controlling the effect across different channels. For this reason, we propose SEAL-L (LoRA), which can be viewed as a superset of SEAL-C. In SEAL-L, while the LoRA module is used, the data and training scheme are derived from the SEAL framework. In the main experiments, we additionally report the results of the SEAL-D (DoRA). SEAL-D replaces the LoRA module with the DoRA (Liu et al. (2024b)) module, a recent variant of LoRA. Through experiments, we demonstrate that SEAL-H and SEAL-C represent the core components responsible for quality improvement.

In the case of SEAL-H, the total number of learnable parameters is $LH$ (the number of blocks * the number of heads). In the case of the LongChat-7B model, this amounts to only 1,024 parameters, making it highly efficient. While SEAL-C uses more parameters, *e.g.*, 128K in LongChat-7B, this cost is still affordable, nearly 10 times smaller than SEAL-L. Furthermore, the dataset contains only 50 samples, resulting in the use of fewer than 2 million tokens for adjusting intensity. Moreover, the

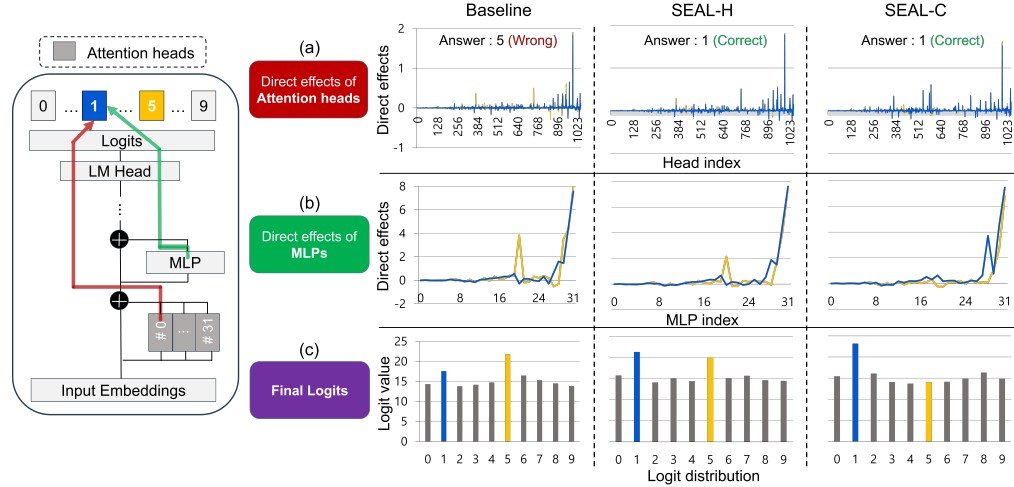

Figure 4: Effects of attention heads and MLP on logits: (a) Direct effects of attention heads, (b) direct effects of MLP layers, and (c) final logits before softmax function for each case. As can be seen from the y-axis scale, the direct effects of MLPs (b) dominated over Attention heads (a).

tuned head-wise or channel-wise scale can be multiplied with the weights of adjacent layers (v_proj or o_proj of Llama) offline, ensuring no additional computational cost during inference time. This efficient design across various aspects highlights the superiority and practicality of SEAL.

# 5 QUALITATIVE ANALYSIS BASED ON DIRECT EFFECT

Before measuring SEAL's performance in downstream tasks, we first conducted a qualitative analysis in this section to provide a deeper understanding of **how the proposed SEAL contributes to improving retrieval scores**. While various circuit analysis techniques have been proposed to analyze the functioning of Transformer architecture, we utilized the direct effect method, which is one of the most intuitive and successful approaches for presenting analysis results. Let $f(p)$ represent the hidden state output of each component (*e.g.*, attention heads, MLPs) for a prompt $p$ whose effect we aim to observe, and we denote the head weight as $W_{head}$. Then the direct effect can be expressed by the following equation:

$$\Delta = W_{head}f(p) \tag{1}$$

Specifically, we utilized a form similar to the direct effect proposed in Lieberum et al. (2023), excluding the normalization term.

## 5.1 DIRECT EFFECT ANALYSIS BEFORE AND AFTER SEAL

For the line retrieval task from the LongEval, we selected an example where the baseline LongChat-7B-32K model produced an incorrect answer, while the tuned model with SEAL provided the correct retrieval answer. The selected example is shown below.

> **Prompt**: ...odd-shrimp: REGISTER_CONTENT is <32616> \nline verdant-efficiency: REGISTER_CONTENT is <24819> \nline permissible-prostanoid:...
> **Question**: Tell me what is the <REGISTER_CONTENT> in line verdant-efficiency? I need the number.
>
> **Correct Answer**: The <REGISTER_CONTENT> in line verdant-efficiency is 24819.
> **Wrong Answer**: The <REGISTER_CONTENT> in line verdant-efficiency is "24856".

We analyzed the impact of each Transformer component on the final logit at the position of the last token in the input, just before the results diverged (1 and 5 in the example above), to examine the role SEAL played in the autoregressive generation process.

The first and second rows of Figure 4 represent the direct effects of all attention heads and MLPs in the models, respectively. In the first row, the multi-heads within the same layer are flattened and

indexed. When comparing the scale of the direct effect metrics, we observed two key findings: first, the influence of the MLPs was more dominant than that of the attention heads. Interestingly, we also identified specific MLPs in the later layers (20th: digit 5, 28th: digit 1) that appeared to amplify the effects on the logits corresponding to the numbers being retrieved.

According to the definition of direct effect, the sum of the direct effects of all components for each token constitutes the final logits, and the difference in this sum leads to variations in retrieval outcomes. In the baseline model, the direct effect of the 20th MLP for the token corresponding to the digit 5 is more dominant than that of the 28th MLP for the digit 1. As a result, this influence is reflected in the logit, leading to the incorrect prediction of the digit 5. However, there is also a peak in the direct effects of MLPs for the correct digit 1, and final logits for the correct answer have the second-highest logit value. This indicates that the baseline model does possess some internal retrieval ability for the correct answer.

In contrast, when examining the direct effects of the MLPs in the proposed SEAL-H model, we observe that the peak value for the digit 5 reduces, while the peak for the digit 1 increases. This is due to the appropriate head-wise scaling of SEAL-H, which eventually influences the final logit and the retrieval results. In the case of SEAL-C, which employs channel-wise scaling, it more precisely scales the effect of attention, resulting in both the direct effect and the logit value clearly favoring digit 1.

Through this, we can understand how SEAL's attention scaling can alter retrieval outcomes. Next, we investigated the quantitative improvements SEAL brings to actual retrieval tasks by evaluating its performance across various down-stream tasks.

## 6 EXPERIMENTAL RESULTS

To validate the effectiveness of the proposed SEAL, we evaluated its retrieval performance on long-context inputs for two widely-used tasks: line retrieval from LongEval and the Needle-in-a-Haystack.

**Models:** We validated SEAL on five models: LongChat-7B-v1.5-32K and Mistral-7B-Instruct-v0.2 (Jiang et al. (2023)), which support a 32K context window length, and Vicuna-7B-v1.5-16K (Chiang et al. (2023)), Vicuna-13B-v1.5-16K, LongChat-13B-16K, which support a 16K context window.

**Settings:** We utilized the Axolotl[1] framework to tune SEAL-H, SEAL-C, SEAL-L, and SEAL-D. The tuning was performed using the AdamW optimizer without learning rate (lr) decay, and all models were tuned for 1 epoch. For tuning in the line retrieval task, SEAL-C used a lr of 2e-2, while SEAL-H used 1e-2 and 2e-2 for the 7B and 13B models, respectively. For the Needle-in-a-Haystack task, learning rates of 4e-2 and 5e-2 were used. For SEAL-L and SEAL-D, LoRA and DoRA modules with $r = 4$ were applied, respectively, to every linear layer in the attention module (QKVO), with a lr of 2e-4. A single A100 80GB GPU was used for both tuning and evaluation.

**Dataset generation:** We used 50 generated samples for each task. Models supporting 32K context window length were tuned with samples containing 31K input tokens, while models supporting 16K context window length used 16K input tokens. For the 7B models, tuning with the 31K dataset took about 40 minutes, and tuning with the 16K dataset took about 10 minutes.

### 6.1 RESULTS ON LINE RETRIEVAL TASK

In Table 1, the baseline models of LongChat and Vicuna show significant score degradation as the input length approaches their context window limits. However, the proposed SEAL methods demonstrate dramatic improvements over the baseline across all input lengths, with particularly notable improvements for LongChat-7B (from 0.32 to 0.88) and Vicuna-13B (from 0.42 to 0.94). Mistral, while not experiencing a steep drop within the 32K input length, also shows substantial improvement in almost all cases, reaching near 100% performance when SEAL is applied.

Compared to SEAL-L (LoRA), which tunes the entire QKVO, SEAL-H achieves comparable performance to LoRA while using approximately 4,000 times fewer parameters. This demonstrates that

---

[1]https://github.com/axolotl-ai-cloud/axolotl

Table 1: Comparison of the line retrieval task scores. Params. (#, %) represent the number of tunable parameters and the ratio of tunable parameters to the total parameters of the baseline, respectively.

| Model | Method | Params. (#, %) | 9K | 14K | 19K | 23K | 28K | 31K |
|---|---|---|---|---|---|---|---|---|
| | Baseline | - | 0.98 | 0.96 | 0.84 | 0.54 | 0.38 | 0.32 |
| | SEAL-H | 1.0K, 1.5e-5% | 1.00 | 1.00 | 0.98 | 1.00 | 0.94 | 0.80 |
| LongChat-7B-v1.5-32K | SEAL-C | 131.1K, 1.9e-3% | 0.98 | 0.96 | 0.94 | 0.92 | 0.94 | 0.88 |
| | SEAL-L | 4.2M, 6.2e-2% | 1.00 | 1.00 | 1.00 | 1.00 | 0.94 | 0.80 |
| | SEAL-D | 4.7M, 7.0e-2% | 1.00 | 1.00 | 1.00 | 1.00 | 0.94 | 0.86 |
| | Baseline | - | 0.98 | 1.00 | 0.90 | 0.86 | 0.88 | 0.94 |
| | SEAL-H | 1.0K, 1.4e-5% | 1.00 | 1.00 | 1.00 | 0.98 | 0.98 | 1.00 |
| Mistral-7B-Instruct-v0.2 | SEAL-C | 131.1K, 1.8e-3% | 1.00 | 1.00 | 1.00 | 1.00 | 1.00 | 0.98 |
| | SEAL-L | 4.2M, 5.8e-2% | 1.00 | 1.00 | 1.00 | 1.00 | 1.00 | 1.00 |
| | SEAL-D | 4.7M, 6.5e-2% | 1.00 | 1.00 | 1.00 | 1.00 | 1.00 | 1.00 |

| Model | Method | Params. (#, %) | 5K | 7K | 9K | 12K | 14K | 16K |
|---|---|---|---|---|---|---|---|---|
| | Baseline | - | 1.00 | 1.00 | 0.96 | 0.92 | 0.60 | 0.64 |
| | SEAL-H | 1.0K, 1.5e-5% | 1.00 | 1.00 | 1.00 | 0.98 | 0.92 | 0.84 |
| Vicuna-7B-v1.5-16K | SEAL-C | 131.1K, 1.9e-3% | 1.00 | 1.00 | 1.00 | 0.94 | 0.96 | 0.98 |
| | SEAL-L | 4.2M, 6.2e-2% | 1.00 | 1.00 | 1.00 | 0.96 | 0.96 | 0.96 |
| | SEAL-D | 4.7M, 7.0e-2% | 1.00 | 1.00 | 1.00 | 0.96 | 0.98 | 0.98 |
| | Baseline | - | 0.96 | 0.94 | 0.92 | 0.92 | 0.80 | 0.60 |
| | SEAL-H | 1.6K, 1.2e-5% | 1.00 | 1.00 | 0.98 | 1.00 | 1.00 | 0.92 |
| LongChat-13B-16K | SEAL-C | 207.7K, 1.6e-3% | 1.00 | 1.00 | 1.00 | 1.00 | 1.00 | 0.96 |
| | SEAL-L | 6.6M, 5.0e-2% | 1.00 | 1.00 | 0.98 | 1.00 | 0.98 | 0.96 |
| | SEAL-D | 7.5M, 5.6e-2% | 1.00 | 1.00 | 0.98 | 1.00 | 0.98 | 0.96 |
| | Baseline | - | 0.98 | 0.98 | 0.94 | 0.88 | 0.68 | 0.42 |
| | SEAL-H | 1.6K, 1.2e-5% | 1.00 | 1.00 | 0.96 | 1.00 | 0.96 | 0.94 |
| Vicuna-13B-v1.5-16K | SEAL-C | 207.7K, 1.6e-3% | 1.00 | 1.00 | 0.96 | 0.98 | 0.98 | 0.94 |
| | SEAL-L | 6.6M, 5.0e-2% | 0.98 | 0.98 | 0.88 | 1.00 | 1.00 | 0.90 |
| | SEAL-D | 7.5M, 5.6e-2% | 1.00 | 0.98 | 0.90 | 1.00 | 1.00 | 0.92 |

tuning the head-wise influence of attention is key to improving retrieval performance, a finding also validated through analysis. Additionally, when comparing SEAL-H to SEAL-C, SEAL-C generally exhibits higher performance, confirming that fine-grained control at the channel-wise level is important, even within the influence of heads. These results support our analysis that varying the strength of each head can significantly enhance long-context retrieval capabilities in a cost-efficient manner.

## 6.2 RESULTS ON NEEDLE-IN-A-HAYSTACK TASK

Figure 5 presents the results of applying SEAL to the Needle-in-a-Haystack task. While Mistral doesn't collapse at longer input than 32K, it still experiences performance degradation with significantly longer inputs. Despite using only 50 samples and training with synthesized needles that are different from the actual target needle, as depicted in Figure 3, SEAL demonstrates remarkable performance improvement. Below are examples of correct and incorrect responses of the LongChat-7B-v1.5-32K model at a length of 20533 tokens, 22% depth of needle insertion.

---

**Prompt**: ...It's a worrying prospect. The best thing to do in San Francisco is eat a sandwich and sit in Dolores Park on a sunny day. It would be a bummer to have another grim monoculture like...
**Question**: What is the best thing to do in San Francisco?

**SEAL-C (score: 100%)**: The best thing to do in San Francisco is eat a sandwich and sit in Dolores Park on a sunny day.
**Baseline (score: 8.3%)**: Go to the top of the hill at Lands End and look out at the city.

---

Although SEAL-H shows slightly lower performance than SEAL-C or SEAL-L, it once again confirms that retrieval performance can be greatly recovered by simply adjusting the head-wise influence through scalar values, amounting to only 1024 parameters for the entire 7B model.

Interestingly, in the case of Mistral, even though sample data were generated for a length of 31K for the SEAL method, performance improved with inputs much longer than 31K. However, for LongChat and Vicuna, the naive application of SEAL does not allow them to extend beyond their learned context window length.

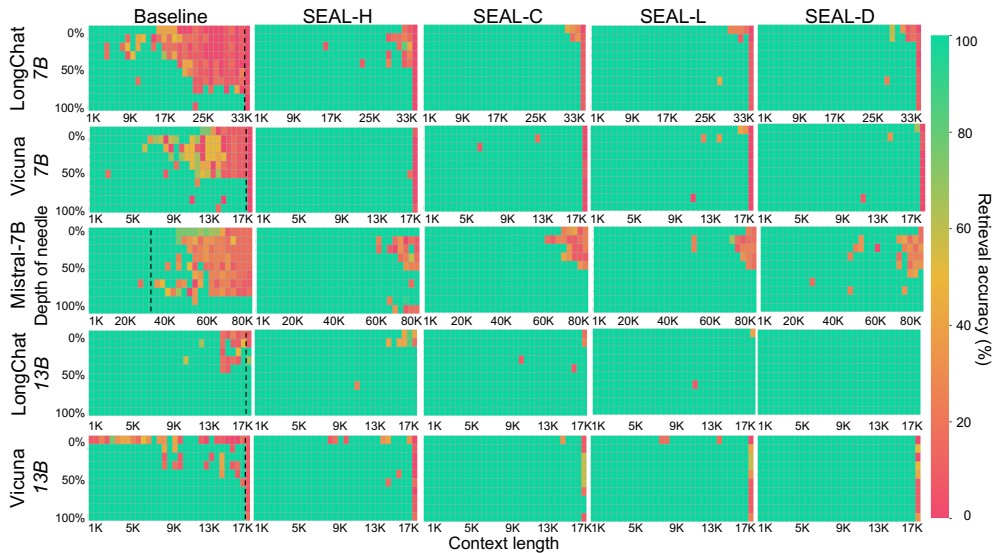

Figure 5: Comparison of Needle-in-a-Haystack performances. The x-axis and y-axis represent the token length and the positions where the needle is inserted, respectively. The dotted black lines denote the context window limits of the original models.

Table 2: Line retrieval task scores for context length extension methods with and without SEAL in Llama-2-7b-Chat.

| Method | 5K | 7K | 9K | 12K | 14K | 16K | Method | 5K | 7K | 9K | 12K | 14K | 16K |
|---|---|---|---|---|---|---|---|---|---|---|---|---|---|
| Baseline | 0.00 | 0.00 | 0.00 | 0.00 | 0.00 | 0.00 | Baseline | 0.00 | 0.00 | 0.00 | 0.00 | 0.00 | 0.00 |
| NTK | 0.88 | 0.32 | 0.16 | 0.00 | 0.00 | 0.00 | Self-Extend | 0.76 | 0.52 | 0.46 | 0.26 | 0.22 | 0.10 |
| +SEAL-C | **0.90** | **0.92** | **0.92** | **0.84** | **0.74** | **0.88** | +SEAL-C | **0.96** | **0.96** | **0.90** | **0.84** | **0.68** | **0.56** |

# 7 SEAL WITH TRAINING-FREE CONTEXT LENGTH EXTENSION

In this work, we address one of the two major problems that can arise with lengthy inputs: the gradual decline in performance within the context window. However, our approach can be used orthogonally to methods that extend the context window length itself. In fact, the application of SEAL to models like LongChat is an example where the Llama (Touvron et al. (2023)) model has already been extended with context windows through RoPE scaling and fine-tuning. However, such tuning-based extensions come with significant costs in terms of time, data, and training infrastructure.

Recently, training-free context length extension methods (*e.g.*, NTK (bloc97 (2023a)), Self-Extend (Jin et al. (2024))) have emerged and garnered considerable attention. However, it is important to note that these methods generally exhibit lower performance compared to fine-tuning-based approaches (*e.g.*, PI (Chen et al. (2023)), YaRN (Peng et al. (2023))). If SEAL could be applied orthogonally to these training-free context length extension methods, it would offer the attractive possibility of simultaneously leveraging the low-cost advantages of the SEAL and tuning-free approach while restoring performance degradation through SEAL.

The results in Table 2 show that when extending the effective context length of Llama-2-7b-Chat to over 16K using only NTK or Self-Extend, the retrieval performance at lengths greater than 8K drops significantly. However, by utilizing SEAL in combination to adjust the attention influence, we can dramatically improve performance beyond the original base model's context window limitation (4K of Llama). Notably, NTK is completely unable to retrieve information at lengths above 12K, yet with the application of SEAL, it achieves performance comparable to that at shorter lengths.

Figure 6 presents the measured performance results for the Needle-in-a-Haystack task, further demonstrating that SEAL significantly enhances the insufficient performance of the training-free context length extension methods. These results enable a practical approach to effectively increase

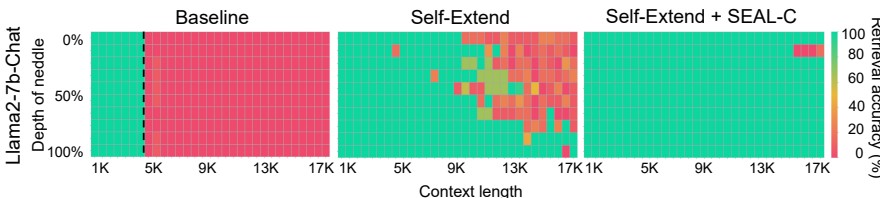

Figure 6: The results of Needle-in-a-Haystack in Llama-2-7b-Chat. The dotted black line denotes the context window limits of the original Llama model: 4k tokens.

Table 3: The retrieval performance of out/in-domain long-context tasks in LongChat-7B-v1.5-32K.

| Domain | Method | Single Doc QA | | | | | Multi Doc QA | | | | |
| | | MultiField QA-EN | MultiField QA-ZH | Narrative QA | Qasper | Avg. | HotPot QA | 2WikiM QA | Musique | DuReader | Avg. |
|---|---|---|---|---|---|---|---|---|---|---|---|
| | Baseline | 42.52 | 35.15 | 20.66 | 29.16 | **31.87** | 33.12 | 23.89 | 14.49 | 21.66 | **23.29** |
| Out-of-domain | SEAL-H | 43.26 | 36.94 | 19.65 | 32.61 | **33.12** | 30.55 | 24.07 | 15.67 | 24.22 | **23.63** |
| | SEAL-C | 42.23 | 37.57 | 20.26 | 31.77 | **32.96** | 32.55 | 23.85 | 13.34 | 24.37 | **23.53** |
| In-domain | SEAL-H | 41.46 | 36.57 | 20.21 | 35.82 | **33.52** | 38.85 | 23.13 | 19.24 | 23.71 | **26.23** |
| | SEAL-C | 44.02 | 43.35 | 19.59 | 34.86 | **35.46** | 45.13 | 32.50 | 22.93 | 24.52 | **31.27** |

the context length of any model at less than 1% of the cost associated with fine-tuning-based context length extension methods by combining training-free context length extension with SEAL.

## 8 GENERALIZATION ABILITY OF SEAL

The proposed SEAL method adopts a task-specific approach using formatted data for particular downstream tasks, but it is fundamentally based on the theoretical premise of scaling attention components to enhance retrieval capabilities. To evaluate whether SEAL can deliver general improvements in retrieval performance for out-of-domain tasks, we measured the scores for the QA task type in LongBench using the scaling values learned from the line retrieval task in Section 6.1. We used the learned scaling values of the LongChat-7B model, which showed the largest performance improvement in line retrieval. We also provided results when LongBench was evaluated as an in-domain manner. Additionally, to ensure that SEAL's retrieval-focused scaling does not degrade the inherent knowledge or reasoning abilities of the LLMs, we measured the MMLU (Hendrycks et al. (2020)) scores.

When using scale values tuned for the line retrieval task, the out-of-domain MMLU results are 42.53 / 42.34 / 42.17 for baseline, SEAL-H, and SEAL-C, respectively. The MMLU scores remain nearly unchanged, indicating that our method effectively identifies and scales only the attention heads relevant to long-context retrieval. Additionally, despite SEAL being applied task-specifically to line retrieval, which focuses on retrieving numbers, Table 3 shows that the scores in the out-of-domain LongBench metrics are maintained or even slightly improved. This demonstrates that the retrieval performance gains achieved by SEAL contribute to tasks like document QA, confirming the generalization capability of our approach.

## 9 CONCLUSION

The ability to retrieve and extract information from long-length input is an important component of the LLMs. Through our analysis, we found that there are attention heads that have a good or bad impact on the retrieval scores. Based on this, we introduce SEAL, a cost-efficient attention strength scaling method to deliberately control the impact of each head. Despite using very few formatted sample data and scaling parameters, SEAL maintains generalization performance and significantly improves retrieval performance. We believe that our insights will promote the widespread adoption of LLMs.

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

## A  Training Configurations

**LongBench in-domain tuning**

We used the same training hyperparameters with line retrieval fine-tuning. In specific, we used learning rate 1e-2 and 2e-2 for SEAL-H and SEAL-C, respectively.

**SEAL with Training-free context length extension** For NTK, we set the scaling factor to 4 to extend the context length from 4096 to 16384. For Self-Extend, we set the group size to 6 and the neighbor window size to 1024 to extend the length to (4096-1024) × 6 = 18432.

We used a learning rate of 4e-4 for the tuning SEAL-C of Mistral, on the Needle-in-a-Haystack samples.

## B  Generating sample data for downstream task

### B.1  Line retrieval

LongEval provides generate_testcases.py to create random data of the desired length. We created a prompt (input) for the sample utilizing that code. The answer label for scale tuning is made as follows:

```
data['answer_str']  =  f"The  <REGISTER_CONTENT>  in  line  data['random_idx'][0]  is
data['expected_number']."
```

We further used appropriate system prompts and conversation templates for each model when training with axolotl.

### B.2  Needle-in-a-Haystack

The pipeline for generating sample data for Needle-in-a-Haystack is detailed in the Figure 3. We used the following input prompt to generate random needles using chatGPT:

```
I am trying to test the retrieval performance of the model. I need needle sentences to find in a long
context, with the corresponding retrieval question. Here is one example case: "needle": "The first thing
you notice upon entering the room is the bright green chair sitting in the center facing the window.",
"question": "What is the first thing you notice upon entering the room?". I want to make 10 sets of nee-
dles and corresponding retrieval questions in jsonl format, like "needle": "...", "question": "...". Here are
some restrictions about needles and questions. 1. Since the purpose is to test only retrieval performance,
the less it is related to general knowledge, the better. 2. It is better to place the content corresponding to
the question at the beginning of the needle sentence, like the given example. 3. Keep the length of the
needle similar to or longer than the length of the example needle provided. 4. Please give variations to
the format, "first thing".
Can you make 10 sets of examples for me?
```

The 10 random needle and question pairs created from the above prompt are as follows:

"needle": "Immediately noticeable upon entering the room is the large oak table positioned beneath the chandelier.", "question": "What is immediately noticeable upon entering the room?"
"needle": "A striking feature of the room is the tall bookshelf that spans the entire length of the far wall.", "question": "What is a striking feature of the room?"
"needle": "Dominating the center of the room is a grand piano, its polished surface reflecting the light from the windows.", "question": "What dominates the center of the room?"
"needle": "Catching your eye as you step inside is the intricate tapestry hanging on the left wall, its colors vivid and bright.", "question": "What catches your eye as you step inside?"
"needle": "The first thing that draws your attention is the large framed photograph resting on the mantel.", "question": "What is the first thing that draws your attention?"
"needle": "Clearly visible as you enter is the large circular rug that covers most of the hardwood floor.", "question": "What is clearly visible as you enter?"
"needle": "What stands out immediately is the tall standing lamp positioned next to the armchair in the corner.", "question": "What stands out immediately in the room?"
"needle": "The most noticeable item upon stepping inside is the antique grandfather clock, ticking rhythmically in the corner.", "question": "What is the most noticeable item upon stepping inside?"
"needle": "Your attention is immediately drawn to the stained glass window, casting colorful patterns of light across the floor.", "question": "What is your attention immediately drawn to?"
"needle": "Visible as soon as you enter the room is a large painting of a landscape, mounted prominently on the main wall.", "question": "What is visible as soon as you enter the room?"

## C ANALYSIS ON NUMBER OF SAMPLES AND LEARNING RATE

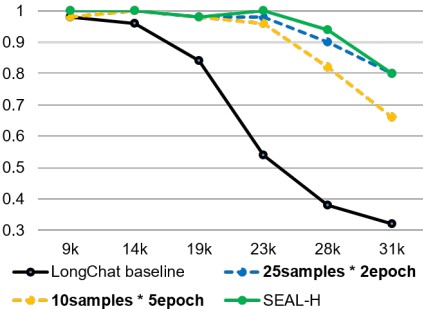

Figure 7: Line retrieval results when using fewer samples than the default 50 samples.

One of the advantages of SEAL is that it can achieve significant performance improvements with a very small number of formatted data samples. To analyze the impact of the number of samples on scale tuning, as well as the influence of the key hyperparameter, learning rate. We tuned the scale of SEAL-H by sweeping the learning rate and the number of samples. For this experiment, we generated a new set of 100 random samples for line retrieval using the same method proposed in Appendix B. The results of applying SEAL-H to LongChat-7B-v1.5-32k with different hyperparameter configurations are shown in Table 4. Generally, performance improves as the number of samples increases, and for LongChat, a learning rate of 3e-2 was identified as the best configuration. However, for general configurations, we adopted a learning rate of 1e-2 in the main experiments.

Additionally, we tested whether comparable performance improvements could be achieved using significantly fewer samples, with only 25 or 10 samples. In Figure 7, we compared tuning with 25 samples over 2 epochs and 10 samples over 5 epochs against the original SEAL-H (which used 50

|      | 10   | 30   | 50   | 70   | 99   |
|------|------|------|------|------|------|
| 5e-3 | 0.56 | 0.64 | 0.68 | 0.70 | 0.72 |
| 1e-2 | 0.68 | 0.74 | 0.78 | 0.82 | 0.82 |
| 2e-2 | 0.70 | 0.82 | 0.82 | 0.84 | 0.70 |
| 3e-2 | 0.76 | 0.84 | 0.90 | 0.86 | 0.82 |

Table 4: 31k line retrieval results on LongChat, with different learning rates (y-axis) and the number of samples (x-axis).

samples). The results show that even with as few as 25 samples, it is possible to achieve comparable performance. Although there is a relative performance decrease when tuning with only 10 samples for 5 epochs, it is remarkable that even with just 10 samples, there is a substantial improvement over the baseline. Preparing around 10 samples can be easily done by hand without the need for a complex data processing pipeline, which highlights the cost-effectiveness of the SEAL method.

