# OpenReview forum: "SEAL: Scaling to Emphasize Attention for Long-Context Retrieval"
_ICLR.cc/2025/Conference — Submitted to ICLR 2025_

### Official Review · Reviewer_Xi5C · 2024-10-16

**Soundness:** 3
**Presentation:** 3
**Contribution:** 3
**Rating:** 6
**Confidence:** 4

**Summary:**

This paper proposes a novel and practical method, SEAL, to improve the long-context retrieval ability of LLMs.

First, through perturbation experiments, it finds a certain attention head or a certain channel in it can cause a positive or negative effect on long-context retrieval accuracy.

Second, it demonstrates directly scaling the hidden states of these heads or channels can indeed improve the retrieval accuracy of LLMs.

Third, it adds trainable scale factors into the model and use a small amount of samples of retrieval tasks to fine-tune the model. The results show SEAL can remarkably improve the long-context retrieval ability of LLMs.

**Strengths:**

1. This paper discovers that a certain attention head can cast a remarkable positive or negative effect on long-context retrieval accuracy, even as well as a certain channel. This is interesting and helpful for us to further understand the role of the internal modules of LLMs.

2. The proposed method, SEAL, is very cost-effective, which only needs very few training samples and tuned parameters.

3. There are enough evaluation results of various models to demonstrate the method’s effect.

**Weaknesses:**

1. Narrow scope

The method seems to only be applicable for classic retrieval tasks such as NIAH, and the training data is also the same types of tasks. It will not be surprising that this leads to an improvement, since this task has been too simple, fixed and formulaic, which may represent a narrow application scope for this method. It would be better to train and test on more tasks such as Knowledge-QA.

2. No unique advantages

The author should empirically test whether the time or space required by SEAL is significantly less than that of LoRA. Otherwise it cannot show significant superiority of SEAL compared to LoRA. Because the parameters tuned by LoRA are already very few. Though SEAL can theoretically tune much less parameters, it may not significantly save much time.

3. There is little detailed description about the procedures of the method in the abstract or introduction. This will make it hard for readers hard to grasp the method quickly. There usually should be a paragraph included in the introduction to describe the specific operation of the method.

4. The curve of data points in Figure 4 (a) may be too small, making it hard to clearly see the changes.

**Questions:**

1. Can you demonstrate the unique advantages of your method compared to LoRA through more experiments?
2. Can you train and test on more various task types to demonstrate the generalization of your method?

---

> ### Author Response · Authors · 2024-11-23
> **Response to reviewer Xi5C**
>
> ## Generalization capability of SEAL
> Regarding the generalization of SEAL discussed in Section 8, we conducted additional experiments on the more complex Knowledge-QA task type from LongBench. We kindly ask you to refer to the results, which demonstrate that SEAL consistently improves performance in both Single Doc QA and the more complicated Multi Doc QA tasks.
>
> Specifically, the in-domain results in Table 3 were obtained by tuning the scale with format information from certain sub-tasks of LongBench. These results exhibit significant average score improvements across QA tasks, demonstrating that SEAL performs well even on more complex tasks and within a broader scope.
>
> Regarding your concern about "training data is also the same types of tasks," the out-of-domain results in Table 3 address cases where this is not true. For these experiments, the attention scale trained on the format of the Line retrieval (retrieving numerical data) was applied to the LongBench tasks. Despite LongBench being an English-based QA task, we observed slight average score improvements. This finding highlights that SEAL remains applicable to out-of-domain tasks through enhanced retrieval performance, further showcasing its generalization capability.
>
> Moreover, in addition to the Documentation QA results we reported, we are conducting experiments on more diverse task types as requested by the reviewer. However, due to the time constraints of the rebuttal process, we are submitting the remainder of our responses first. If experiments on additional tasks are completed during the rebuttal period, we will provide an updated response accordingly.
>
> ## Advantages of SEAL and comparison to PEFT
> By comparing SEAL-L, SEAL-H, and SEAL-C, we empirically validated that we accurately identified the core components of retrieval and devised the optimal method to update them, ultimately proposing a cost-effective fine-tuning technique.
>
> Please note that our primary focus, as mentioned in the global response, is not on proposing a more efficient PEFT method but rather on introducing the SEAL pipeline, analyzing its impact on retrieval performance, and identifying key components through this process.
>
> Also, our SEAL pipeline is a solution that is not bound to any specific PEFT method. We hope it is recognized that SEAL-H, C, L (LoRA), and D (DoRA) are all approaches initially proposed by us.
>
> We acknowledge that our contributions might not have been clearly conveyed, which may have led to the misconception that this paper is about proposing an efficient PEFT technique. To address this, we have revised the manuscript's flow and tone to improve the overall writing quality. We hope this is taken into account.
>
> ## Description about the procedures of SEAL
> Thank you for your insightful comment. We have added a brief description of the procedures of SEAL to Line 77 of the introduction section of the revised manuscript, which is written in blue color. We invite you to review this updated content in the revised manuscript.
>
> ## Improving visibility in Figure 4
> We have revised the relevant content in Figure 4 of the updated manuscript, where the modified part is marked with a blue box. Our initial intent was to align the y-axis scales of Figures 4(a) and 4(b) to emphasize that the direct effect of MLPs dominates over Attention Heads. We have modified the caption to preserve this intent while ensuring that the changes are more clearly observable.

---

> ### Comment · Reviewer_Xi5C · 2024-11-23
>
> Your reply makes the advantage of SEAL clearer. I would keep my overall score, but improve the score of contribution.

---

> ### Author Response · Authors · 2024-12-04
> **Response to reviewer Xi5C**
>
> In response to the reviewer's request, we conducted additional experiments on the tasks of the latest and more refined benchmark, LV-Eval [1]. We apologize for the delayed response.
>
> We reviewed various long-context benchmarks, such as ZeroSCROLLS [2] and L-Eval [3], and found that many of the datasets in these benchmarks (e.g., Qasper, MuSiQue, NarrativeQA) already overlap with the datasets of LongBench [4], which we had already experimented on. Therefore, we selected some sub-tasks from the more recent LV-Eval benchmark to minimize overlap with those in LongBench.
>
> Importantly, to address the reviewer's two concerns: **(1)** SEAL is only applicable for classic retrieval tasks like Needle-in-a-Haystack and **(2)** the training data is also the same types of tasks, we used the SEAL-H scales trained on the data from triviaqa_e sub-task of LongBench, which was used in the experiments in Section 8 of our paper.
>
> The results are as follows:
>
> | LongChat-7B-v1.5-32K      | input length   | loogle-SD-mixup | cmrc-mixup | loogle-CR-mixup | Average   |
> |------|----|----|----|----|----|
> | **Baseline** | 16K    | 27.42  | 20.99   | 11.25  | 19.89  |
> |              | 32K   | 18.21   | 10.77   | 11.17  | 13.38  |
> | **SEAL-H**  | 16K    | 30.08  | 22.62  | 11.69   | **21.46**  |
> |              | 32K    | 20.44 | 12.19  | 12.48   | **15.04**  |
>
> loogle-SD-mixup and cmrc-mixup tasks are Single-hop QA tasks. loogle-CR-mixup task is Multi-hop QA task.
>
>
> Even though we used the head-wise scale trained specifically for LongBench's triviaqa_e sub-task (which differs entirely from LV-Eval’s sub-tasks), we observed performance improvements at both the 16K and 32K context lengths across many tasks.
>
> This demonstrates two key points:
>
> 1. **Robust Performance on Challenging Benchmarks:** SEAL performs well even on the more challenging LV-Eval benchmark, which consists of single-hop QA tasks and multi-hop QA tasks. This could be the answer to concern **(1)** above.
> 2. **Generalization Capability:** The improved retrieval performance achieved through SEAL extends to other QA tasks, even when using head-wise scales fine-tuned on the task format from different benchmark. This highlights SEAL's strong generalization capability. This could be the answer to concern **(2)** above.
>
> We hope these experimental results will assist the reviewer and the Area Chair in their evaluation.
>
>
> **References:**
>
> [1] LV-Eval: A Balanced Long-Context Benchmark with 5 Length Levels Up to 256K
>
> [2] ZeroSCROLLS: A Zero-Shot Benchmark for Long Text Understanding
>
> [3] L-Eval: Instituting Standardized Evaluation for Long Context Language Models
>
> [4] LongBench: A Bilingual, Multitask Benchmark for Long Context Understanding

---

### Official Review · Reviewer_K91h · 2024-10-20

**Soundness:** 3
**Presentation:** 3
**Contribution:** 2
**Rating:** 5
**Confidence:** 4

**Summary:**

This paper proposes an approach called Scaling to Emphasize Attention for Long-context retrieval (SEAL), which emphasizes specific heads or channels (attention outputs) particularly related to long-context retrieval by efficiently adjusting the strength of each attention component. The authors claimed that SEAL achieves significant improvements in in-domain retrieval performance and cross-domain document QA tasks, also extends the context limits of LLMs while maintaining highly reliable outputs.

**Strengths:**

1. This paper proposes SEAL to efficiently adjusting the strength of each attention component, and achieves superior performance to various LLM baselines in long-context retrieval.
2. The content, figures, and tables of the paper provide a detailed explanation and analysis of the motivation, methods, and experiments, facilitating the readers' understanding.

**Weaknesses:**

1. The experimental results in Table 1 show that SEAL-H and SEAL-C require fewer parameters than Baseline and SEAL-L. However, their performance does not consistently surpass SEAL-L in long-context scenarios, failing to demonstrate the authors' claims.
2. The experiments only select SEAL-L as the baseline, it should include other PEFT methods for comparison.

**Questions:**

1. Except fewer parameters, what other advantages does SEAL have over LoRA or other PEFT methods? Since the parameters of SEAL-L is also small compared to LLMs, what are the unique application scenarios for SEAL?

---

> ### Author Response · Authors · 2024-11-23
> **Response to reviewer K91h**
>
> ## About SEAL-H, SEAL-C, and SEAL-L
> First, thank you for your valuable feedback. Could you kindly clarify which specific aspects you find to be failing to demonstrate the authors' claims? If you provide more details, we will address them thoroughly and promptly.
>
> Our main claim is that SEAL-H and SEAL-C achieve performance comparable to SEAL-L despite using significantly fewer parameters, thereby demonstrating that the core component for improving long-context retrieval performance lies in the scaling of attention components.
>
> ## Experiments with additional baseline DoRA
> In response to the reviewer’s suggestion, we have reported results using [1] DoRA (ICML 2024), a recent variant of LoRA that has gained substantial attention. The results for DoRA (SEAL-D) have been added to Table 1 and Figure 5 of the updated manuscript, which are marked in blue, and we plan to include a detailed explanation in the revised version.
>
> The experimental results show that the performance of SEAL-H and C is comparable to that of LoRA (SEAL-L) and DoRA (SEAL-D).
> This again highlights that scaling attention heads/channels is a key component in improving long-context retrieval performance.
>
> ## SEAL's key contributions and the relationship to PEFT
> By comparing SEAL-L, SEAL-H, and SEAL-C, we empirically validated that we accurately identified the core components of retrieval and devised the optimal method to update them, ultimately proposing a cost-effective fine-tuning technique.
>
> Please note that our primary focus, as mentioned in the global response, is not on proposing a more efficient PEFT method but rather on introducing the SEAL pipeline, analyzing its impact on retrieval performance, and identifying key components through this process.
>
> Also, our SEAL pipeline is a solution that is not bound to any specific PEFT method. We hope it is recognized that SEAL-H, C, L (LoRA), and D (DoRA) are all approaches initially proposed by us.
>
> We acknowledge that our contributions might not have been clearly conveyed, which may have led to the misconception that this paper is about proposing an efficient PEFT technique. To address this, we have revised the manuscript's flow and tone to improve the overall writing quality. We hope this is taken into account.
>
> **References:**
>
> [1] DoRA: Weight-Decomposed Low-Rank Adaptation

---

> > ### Comment · Reviewer_K91h · 2024-11-25
> >
> > Thanks you for the responses, but I'm not sure if my concerns have been addressed.
> >
> > 1. You mentioned that "Our main claim is that SEAL-H and SEAL-C achieve performance comparable to SEAL-L despite using significantly fewer parameters, thereby demonstrating that the core component for improving long-context retrieval performance lies in the scaling of attention components" in your response. I understand that SEAL is still an efficient PEFT method but emphasizing specific attention components through learning, to enhance the performance of LLMs in long-context (l<d) retrieval.
> >
> > 2. From the understanding of point 1, SEAL should be compared with the original LoRA, DoRA or other PEFT methods, to demonstrate the effectiveness and efficiency of SEAL.

---

> > > ### Author Response · Authors · 2024-11-25
> > > **Response to reviewer K91h**
> > >
> > > Thank you for your valuable comments. Here is the summary of our discussion points.
> > >
> > > We are the first to demonstrate that LLMs already contain the necessary information to handle long-context retrieval tasks, and that performance can be significantly improved with minimal adjustments.
> > > However, even for slight adjustments, several design considerations are required:
> > >
> > > **1. Which parameters/adapters should be trained?**
> > >
> > > **2. How to construct the dataset?**
> > >
> > > **3. How much cost should be allocated for training iterations/epochs? (considering overfitting/underfitting issues)**
> > >
> > > We are the first to address the above design points. SEAL represents a comprehensive framework that integrates all three above design points: dataset construction methodology, a tunable parameter space composed of core components, and a training recipe.
> > >
> > > As we emphasized throughout, through analysis of SEAL's impact on retrieval performance, we identified attention components (heads and channels) as the minimal and core components for the first design point.
> > >
> > > Given the absence of a baseline, even when experimenting with LoRA, it should be noted that LoRA is merely one case of adapter design.
> > >
> > > Thus, it serves as a control group specifically addressing the first design point: **determining which parameter/adapter to train.** Consequently, we reported SEAL-L and SEAL-D as experiments where only the first design point was replaced with LoRA and DoRA, respectively.
> > >
> > > **Therefore, we have already conducted comparisons with LoRA and DoRA.** Extending this to modify the second or third design points would go beyond simply using LoRA as a control group and fall outside the scope of our intent.
> > >
> > > Notably, even when the first design point is replaced with LoRA (SEAL-L) or DoRA (SEAL-D), significant performance improvements are observed. This underscores the effectiveness of our remaining second and third design points, as well as the SEAL pipeline as a whole.
> > >
> > > As repeatedly stated, the performance differences between SEAL-L, SEAL-D, and SEAL-H/C are minimal and remain comparable. This reaffirms the efficiency of SEAL-H and SEAL-C, which identify attention components as the core components for the first design point.

---

> > > > ### Comment · Reviewer_K91h · 2024-12-03
> > > >
> > > > Thanks for the authors' responses, which I've read carefully, and I will discuss with other reviewers and AC in the next round.

---

### Official Review · Reviewer_xQUS · 2024-10-22

**Soundness:** 2
**Presentation:** 2
**Contribution:** 3
**Rating:** 5
**Confidence:** 4

**Summary:**

The paper introduces SEAL (Scaling to Emphasize Attention for Long-context retrieval), a novel attention scaling approach that improves retrieval performance for long-context tasks in Large Language Models (LLMs). It addresses the challenge of performance degradation over extended contexts, particularly in retrieval tasks. SEAL fine-tunes specific attention heads or channels using a minimal amount of training data, leading to significant improvements in long-context retrieval across various benchmarks. The paper focuses on cost-efficient enhancement of long-context capabilities without altering the model’s learned behavior.

**Strengths:**

1. SEAL presents an innovative approach by leveraging attention head/channel scaling to enhance long-context retrieval.
2. The method uses very few trainable parameters and requires minimal training data, making it highly efficient.

**Weaknesses:**

1. The term “long-context retrieval” is ambiguous. It would be clearer to refer to “retrieval tasks that have long contexts,” which directly emphasizes tasks like passage retrieval or number retrieval.
2. The paper lacks explicit detail about which context extension techniques are used. For example, Figure 6 mentions the use of Self-Extend, but no experiments isolating its performance are provided.
3. Logical Flow in Writing: Certain parts of the paper are difficult to follow due to writing issues such as ambiguous expressions, inconsistent time tense, and occasional typographical errors (e.g., “biases” instead of “bias”).
4. The distinction between “in-domain” and “out-of-domain” in the experiments is confusing. Specifically, if “in-domain” refers to training on retrieval tasks, why are the same datasets used for both “in-domain” and “out-of-domain” experiments?

**Questions:**

1. What specifically constitutes “long-context retrieval”? Could the authors clarify this definition and provide more precise terminology?
2. Why are different LLMs used in Figures 5 and 6? Is there a specific reason for the model changes, and how do these variations impact the comparability of the results?
3. Can the authors provide experiments isolating the effect of Self-Extend in Figure 5 to verify its individual impact on performance?
4. What is the rationale behind using the same datasets for “in-domain” and “out-of-domain” experiments in Table 3? How is “out-of-domain” defined in this context, and what criteria differentiate the two?

---

> ### Author Response · Authors · 2024-11-23
> **Response to reviewer xQUS (Part 1)**
>
> ## Explanation of the term “long-context retrieval”
> Thank you for your comment. We define the term "long-context retrieval" as performing retrieval tasks under long-context inputs.
>
> Also, in the problem we address, the definition of "long-context" is as follows, as explained in the global response: if the model's context window size is
> $d$ and the input sequence length is $l$, then long-context refers to a sequence length $l$ where $l<d$ but $l\approx d$.
>
> We agree that a precise definition is necessary, given that this is the first attempt to address this problem. Accordingly, we have added this definition to line 67 of the revised manuscript.
>
> ## Overall experimental design and use of context extension techniques
> In the experiments up to Section 6, no additional context extension techniques were applied. We reported the results of the publicly available long-context models (baseline) and the baseline models enhanced with SEAL. Please note that even though the models we used are designed exclusively for long-context tasks, their performance degrades even when the given context is within the model's limitations ($l < d$). When we apply our technique, SEAL, it shows notable improvement across all context lengths. Therefore, the performance improvements compared to the baseline are solely attributed to SEAL, and isolating experiments for other techniques cannot be conducted.
>
> Figure 6 referred to by the reviewer corresponds to the results in Section 7 (SEAL with training-free context length extension), where NTK and Self-Extend were adopted as training-free context extension methods. These methods were applied exclusively in Section 7. This section demonstrates that SEAL can orthogonally contribute to performance improvement when combined with training-free context extension methods for the case of short-context window models + training-free context extension methods + SEAL.
>
> Therefore, the middle subfigure in Figure 6 and the second row of results in Table 2 (labeled NTK and Self-Extend) isolate and evaluate the performance of context extension techniques (NTK and Self-Extend). Once again, no additional context extension methods were applied to the baseline to any of the figures or tables apart from Figure 6 and Table 2.
>
> ## Logical flow in writing
> We have revised not only the parts you pointed out but also improved the overall claim to ensure a more natural logical flow throughout the manuscript. We kindly request you to review the updated version of the manuscript.
>
> ## “in-domain” and “out-of-domain” experiments in Section 8
> When considering tasks A and B, along with their respective format datasets
> a and b, we define in-domain as the scenario where the scaling factor tuned on dataset
> a (or b) is applied to task A (or task B). Conversely, out-of-domain refers to situations where the scaling factor tuned on dataset a is applied to a new task, B, or vice versa.
>
> In our evaluation of the generalization ability of SEAL (Section 8), we designated Line-retrieval and LongBench as tasks A and B, respectively. For the out-of-domain experiments in Table 3, we tuned the scaling factor using the line-retrieval format dataset (a) and tested it on LongBench (task B). This represents an out-of-domain scenario. Even under such conditions, SEAL improves retrieval performance, demonstrating slight improvements in the LongBench average score, despite it being a completely different English QA task.
>
> On the other hand, the in-domain experiments in Table 3 involved tuning the scaling factor using the format data from a subset of LongBench subtasks (b) and testing it on LongBench (task B). This in-domain scenario shows significant accuracy improvements.
>
> Thus, in-domain and out-of-domain experiments represent distinct experimental settings with different training datasets, while only task B is commonly used for verification. Including these results in the same table may have caused some confusion, and we will revise the format to reduce potential misunderstandings.

---

> ### Author Response · Authors · 2024-11-23
> **Response to reviewer xQUS (Part 2)**
>
> ## Using different LLMs in Figures 5 and 6
> Most importantly, the objectives and experimental setups in Section 6 (Figure 5) and Section 7 (Figure 6) are fundamentally different, which is why different LLM families were used.
>
> In Figure 5, the goal was to evaluate the effectiveness of SEAL by applying it to pre-existing long-context models. Therefore, we used open-source models that support relatively large context windows (16k/32k), such as Mistral, LongChat, and Vicuna.
>
> In contrast, Figure 6 aimed to demonstrate that methods for extending the context window and SEAL can be applied orthogonally. For this purpose, we started with Llama-2, a representative foundation model with a short-context window (4k), and applied other context extension methods (NTK, Self-Extend) alongside SEAL. Consequently, the model families evaluated in Figures 5 and 6 have to be different.
>
> Since the evaluation categories in Figures 5 and 6 are distinct, direct comparison of results is not feasible. However, conducting experiments across such diverse configurations and scenarios demonstrates that SEAL performs well regardless of how long-context models are created (e.g., pre-training for Mistral, fine-tuning for LongChat and Vicuna, and training-free extension for Llama + NTK or Self-Extend) or the model family (e.g., Mistral and Llama). This highlights SEAL’s superiority and its strong generalization capabilities.
>
> ## Isolating the impact of context extension methods
> In the experiments for Figure 5 (Section 6), we did not employ any additional context length extension techniques, including Self-Extend. The experiments up to Section 6 focused on applying SEAL to publicly available long-context models (with $d$=16k or 32k) to isolate and evaluate the pure impact of SEAL compared to the baseline. For further details, we kindly ask you to refer to the response provided above (Overall experimental design and use of context extension techniques).
>
> ## The dataset setup and significance of the experiments in Table 3
> The reason LongBench tasks were used in both the in-domain and out-of-domain experiments is as follows:
>
> Out-of-domain experiments: LongBench, being an English QA task, significantly differs in characteristics from line retrieval, which involves simple retrieval of numerical data. Therefore, LongBench was chosen as the task B case for the out-of-domain experiments.
>
> In-domain experiments: Since LongBench and line-retrieval belong to entirely different domains, the performance improvement in the out-of-domain experiments was relatively small. We were concerned that readers might misunderstand this result and assume that SEAL does not effectively enhance retrieval performance for more complex tasks, such as QA. To address this, we added in-domain experiments using LongBench itself.
>
> As explained in previous responses, the in-domain and out-of-domain experiments are distinct, with only the task for the verification (LongBench) being the same. The important point is, the datasets used to tune the scaling factors are different, line-retrieval format data (a) for the out-of-domain case and LongBench format data (b) for the in-domain case. Therefore, we believe there is no issue with the fairness of the experimental setup.

---

> > ### Comment · Reviewer_xQUS · 2024-11-25
> > **Response to the authors' serious concern.**
> >
> > Firstly, I would like to express my apologies for not engaging in the discussion in a timely manner. I understand the frustration expressed by the authors and appreciate their serious concerns. I would like to humbly clarify several points:
> >
> > - The authors suggest that the presentation of the paper should not be a major factor in the assessment for rejection. However, I hold a contrasting opinion: good presentation is a fundamental requirement for any academic paper. After reviewing the revised version, I still noticed several issues, such as typos, inconsistent expressions, and terms being used before they are properly defined. Therefore, I kindly encourage the authors to conduct a thorough inspection of the paper’s writing.
> >
> > - The authors claimed that “the reviewer did not make an effort to understand the paper or its contributions.” I respectfully disagree, as I thoroughly read the paper at least twice before providing my comments. While misunderstandings are common during the review process, I sincerely hope the authors can acknowledge that my critique was made in good faith, based on my understanding of the paper.
> >
> > - Upon re-reading the paper, I recall that my primary concern during the initial review was the “lost-in-the-middle” issue, which the paper claims to address. However, this issue appears to have been largely mitigated by the release of several new LLMs. For instance, many recent models with context lengths of 128K have reported strong performance in needle-in-a-haystack experiments. As a result, I am uncertain whether the impact claimed in this paper will remain significant given the increasing capabilities of newer LLMs, or if the claim that “this breakthrough opens up new possibilities for enhancing the long-context retrieval capabilities of existing LLMs” still holds. I apologize for not mentioning this concern in my initial comments.
> >
> > - The authors expressed serious concern that I “ultimately assign an unreasonably low score that effectively blocks our paper from acceptance.” I sincerely apologize if my scoring caused any discomfort. However, after re-reading the revised paper, I still lean towards rejecting it. Given that three out of four reviewers currently lean towards rejection, I am uncertain if increasing my score from an “unreasonably low score” to a “reasonably low score” (e.g., 5) would change the outcome.
> >
> > I deeply regret any discomfort caused by my review comments. I strongly encourage the authors to remain calm, focus on improving their method and presentation, and consider resubmitting to another venue if the final decision is negative. I wish the authors success in their future submissions.

---

> > > ### Comment · Reviewer_xQUS · 2024-11-25
> > >
> > > Upon receiving the email notification highlighting the authors' serious concerns, I began reviewing their concerns, responses, and the revised paper. However, after commenting, I noticed that their serious concerns were no longer visible to me. I am uncertain whether the authors adjusted the visibility scope or deleted their comments. Both scenarios are quite frustrating for me, as I sincerely wish to engage in direct and constructive discussions with the authors.

---

> > > > ### Author Response · Authors · 2024-11-25
> > > > **Thank you for your renewed comment.**
> > > >
> > > > We acknowledge that our initial tone may have been too strong, so we temporarily adjusted the option. However, after noticing your feedback, we make it visible again. While you mentioned reviewing the manuscript multiple times before providing your comments, we think the initial feedback lacked the level of detail and constructiveness present in your more recent comments. The differences between your initial and current feedback remain evident.
> > > >
> > > > That said, we appreciate the additional insights you’ve provided. Regardless of how your evaluation impacts our score or the acceptance of our paper, our focus has always been on promoting a constructive review process. This process has highlighted areas where we can improve, and we are genuinely grateful for the opportunity to learn and grow from the reviewer's critique. Moving forward, we will address these weaknesses with greater precision in our research. Even if we may have seemed impolite at times, we value the constructive nature of your feedback and wish to express our sincere thanks.

---

> > > ### Author Response · Authors · 2024-11-27
> > > **Response to reviewer xQUS (revised manuscript)**
> > >
> > > Through the reviewer's two responses, we have understood that the primary basis of the evaluation lies in writing quality, including good presentation. Accordingly, we have uploaded a revised PDF that has been further refined from the previous version. We would like to express our sincere gratitude for the suggestions aimed at improving the writing quality of our manuscript.
> > >
> > > Specifically, corrections for grammatical errors such as typos and inconsistent time tenses are marked in red. Other adjustments, including ambiguous expressions, inconsistent terminology, and terms being introduced before proper definitions are marked in blue.
> > > Representative modifications for each category are listed below, and additional revisions have been made throughout the manuscript.
> > >
> > > Ambiguous expressions:
> > > - Revised the description of out-of-domain tasks in the abstract, which was previously referred to as cross-domain.
> > > - Adjusted the mention of "lost in the middle" in Line 66.
> > > - Replaced "full model" with "baseline model".
> > > - Clarified "are derived from the SEAL framework" in Line 263.
> > >
> > > Inconsistent expressions:
> > > - Most importantly, we resolved inconsistencies in the use of the hyphen in "long-context".
> > > - Terms such as "training", "tuning", "fine-tuning", and "learning" were carefully chosen based on context to avoid monotony while ensuring appropriateness.
> > > - Unified the use of "k" (e.g., 31k) to uppercase "K" throughout the manuscript.
> > >
> > > Terms introduced before proper definition:
> > > - Added definitions for SEAL-H and SEAL-C in Line 80.
> > > - Added citations for positional encoding in Lines 55 and 105.
> > > - Included the definition of LxHy in the caption of Figure 2.
> > > - Added a citation for the MMLU task in Line 519.
> > >
> > > We acknowledge that our manuscript may still be imperfect. If any severe errors remain, we would greatly appreciate it if you could inform us, so we can address them immediately. Such feedback would greatly contribute to the future development of our research.
> > >
> > > We have addressed the reviewer's concerns in the later response regarding writing quality and the "lost in the middle" issue. If our response adequately addresses your concerns, we would sincerely appreciate it if you could consider revising the overall score or providing additional feedback for further improvement.

---

> > > > ### Comment · Reviewer_xQUS · 2024-11-27
> > > >
> > > > Thank you for the authors' response. As the revised paper shows overall improvements, I have updated my assessment accordingly.

---

> ### Author Response · Authors · 2024-11-25
> **Concern about the review from xQUS.**
>
> We are writing to raise a serious concern about one of the reviews we received for our submission. We are addressing this issue not only as an author but also as a supporter of this conference, as we believe the review reflects a significant problem with the process.
>
> First, we would like to thank all reviewers for their time and effort. Reviewing papers is very laborious, and we greatly appreciate the constructive feedback from the other reviewers, which will help us improve our work. However, the review in question is both unhelpful and unfair. It provides little more than superficial comments, fails to engage with the core ideas of our research, and ultimately assigns an unreasonably low score that effectively blocks our paper from acceptance.
>
> From the feedback, it is clear that the reviewer did not make an effort to understand the paper or its contributions. If the paper had been properly read, Weaknesses 2 or Question 3 could not have been raised. Instead of providing meaningful critique, the reviewer has cited minor or surface-level issues (e.g., requesting definitions for relatively intuitive terms, and pointing out typographical errors) as the main reasons for assigning a reject rating. We believe that such issues traditionally belong in the suggestion part of a review and are matters that can be adequately addressed during the rebuttal process. However, during the review discussion period, we did not receive any additional questions or engagement from this reviewer. This lack of interaction has denied us the opportunity to clarify misunderstandings or address their concerns.
>
> To be clear, we are not claiming that our paper is without flaws. We fully acknowledge the valid criticisms raised by other reviewers and are working to address them. However, receiving such a dismissive and low-effort review has made us question the fairness of the review process.
>
> We strongly request that this review be reconsidered in the final decision process.  We hope the committee will address this issue and uphold the standards of this esteemed conference.
>
> Thank you for your attention to this matter.

---

> ### Author Response · Authors · 2024-11-25
> **Response to reviewer xQUS**
>
> We have carefully considered the reviewer's follow-up response and greatly value this opportunity to enhance our research through constructive discussion. We also understand the importance of writing as emphasized by the reviewer, and we will make every effort to correct flaws up until the deadline of the rebuttal.
>
> Regarding the concern about the “lost-in-the-middle” issue raised by the reviewer in a subsequent response, we would like to emphasize that SEAL is not limited to addressing this specific issue or tasks such as Needle-in-a-Haystack.
>
> First, from a practical perspective, one of SEAL’s major advantages, as reconfirmed by the reviewer, is "enhancing the long-context retrieval capabilities of existing LLMs”. Many users rely heavily on long-context LLMs they already use due to various reasons (e.g., license, fine-tuned on specific data, or compatibility with inference frameworks).
>
> For instance, if a user has invested significant resources in fine-tuning a Mistral-7B-Instruct model with their custom documents, re-fine-tuning this data on a new 128k model would be prohibitively costly. In contrast, SEAL can improve retrieval performance for various existing long-context models at a minimal cost.
>
> Additionally, the majority of LLMs are short-context models (<8k). SEAL offers an additional unique advantage by enabling these existing or the latest short-context models to be transformed into long-context LLMs with very low training costs, as demonstrated in Section 7.
>
> Second, as models become increasingly sophisticated, benchmarks for testing long-context capabilities are also advancing. In Section 8 (In-domain results of Table 3), we demonstrated that SEAL significantly improves performance on a much more complex task: Documentation QA, which involves reading lengthy documentation and performing Question-Answering. This result highlights that SEAL is not limited to the Needle-in-a-Haystack task but is a scalable technique capable of enhancing retrieval performance even in scenarios where specific long-context models struggle with more challenging benchmarks.
>
> Finally, beyond the performance of LLMs, the interpretability of LLMs is critically important. Through detailed analysis, we have investigated how attention components influence retrieval performance, which could contribute to future research. For example, pre-training long-context model often requires extensive datasets with long sequences and substantial training resources. Our analysis of core components and mechanisms might help make this process more efficient.
>
> We hope this addresses the reviewer’s latest concerns and provides further clarity regarding SEAL’s contribution and impact on a broader scope.

---

### Official Review · Reviewer_FZZr · 2024-10-30

**Soundness:** 2
**Presentation:** 2
**Contribution:** 3
**Rating:** 5
**Confidence:** 4

**Summary:**

This work focuses on scaling to emphasize attention to long-context retrieval, designed to enhance the retrieval performance of LLMs in handling extended contexts. A cost-effective, learning-based mechanism is proposed to improve the model's performance in long-context retrieval tasks, which emphasizes specific attention heads tailored to retrieval tasks. Experimental results demonstrate superior performance over the compared baselines.

**Strengths:**

1. This paper is well-organized and easy to read.
2. The proposed method presents a reasonable approach for long-context retrieval by identifying the key components of Transformer architecture to boost retrieval performance.
3. The approach is practical and has the potential for broad application in various RAG settings.

**Weaknesses:**

1. The term "cost-efficient" is not clearly defined, resulting in ambiguity when assessing the cost-effectiveness of the approach. The strategy of identifying key components initially and subsequently fine-tuning these components may prove to be computationally intensive. It would be beneficial to provide details regarding the computational time involved in this process.
2. A more thorough evaluation would benefit from comparisons with a broader range of advanced baseline models. Currently, the proposed method is compared against only one simple. Including more sophisticated long-context modeling methods and state-of-the-art techniques would better validate the effectiveness of the proposed method.
3. To confirm the versatility of the proposed method, it would be beneficial to conduct experiments on different LLMs of varying sizes.

**Questions:**

Please refer to the weaknesses.

---

> ### Author Response · Authors · 2024-11-23
> **Response to reviewer FZZr**
>
> ## Explanation of "cost-efficiency"
> We apologize for the unclear explanation of cost-efficiency. We define cost-efficient as the absence or minimal occurrence of various overhead costs during training and inference. Specifically, cost-efficiency in our paper includes 1) A very small number of learnable parameters, 2) The ability to fine-tune with as few as 50 samples, and 3) Achieving tuning within an hour using a single GPU for the training phase. Also, no additional overhead is incurred during the inference phase.
>
> As noted in the dataset generation explanation in Section 6, tuning with 50 samples on a 7B model takes approximately 40 minutes for a dataset length of 31k and about 10 minutes for a dataset length of 16k. This demonstrates that our method is highly efficient and practical.
>
> Regarding SEAL, the key component identification and fine-tuning processes are not separate. As described in Section 4.2, we design SEAL based on the observation that each attention component needs to be emphasized or suppressed to adjust the influence of attention components. Based on this observation, we proposed variations of SEAL that exclusively focus on the head-related learnable parameters.
> Thus, no additional time or process is required for component identification.
>
> Please note that our main contribution lies in demonstrating that LLMs inherently possess retrieval capabilities, which can be effectively restored with minimal fine-tuning, and in identifying that attention heads are the key components in this process. We would like to clarify that cost-efficiency is a natural byproduct of our main contributions, rather than cost-efficiency itself is a primary objective.
>
> ## Broader range of advanced baselines
> To the best of our knowledge, our work is the first to improve retrieval performance for long-context that do not exceed the model’s context window. Consequently, we hope you can understand the difficulty of establishing appropriate or fair baselines for comparison. For instance, previous context length extension methods have entirely different objectives and typically require significant additional training costs.
>
> From this perspective, we include SEAL-L as a main baseline to validate the efficiency of our observation in enhancing LLM's performance with a minimal fine-tuning and generated dataset and to explain the differences between SEAL-L and SEAL-H/C, which reveal the effectiveness of head-wise/channel-wise scaling in the context of retrieval enhancement. Please refer to the global response for more details.
>
> In response to the reviewer’s suggestion to include state-of-the-art techniques, we have reported results using [1] DoRA (ICML 2024), a recent variant of LoRA that has gained substantial attention. The results for DoRA (SEAL-D) have been added to Table 1 and Figure 5 of the updated manuscript, and we plan to include a detailed explanation in the revised version.
>
> The experimental results show that the performance of SEAL-H and C is comparable to that of LoRA (SEAL-L) and DoRA (SEAL-D).
> This highlights that scaling attention heads/channels is a key component in improving long-context retrieval performance, further demonstrating the effectiveness of SEAL.
>
> Once again, one of our main contributions is identifying attention heads and channels as the key components in enhancing the long-context retrieval performance inherent in LLMs through minimal fine-tuning. SEAL can be combined with the superset of key components, PEFT methods, as demonstrated by the LoRA and DoRA extension experiments. These experiments illustrate SEAL’s adaptability and potential for broader application across various PEFT techniques.
>
> ## Experiments with larger model sizes
>
> Thank you for the valuable feedback regarding the verification of versatility. In response, we have conducted additional experiments on the 13B models of the LongChat and Vicuna model families (longchat-13b-16k, vicuna-13b-v1.5-16k). The results of the Line retrieval task and Needle-in-a-Haystack task have been added to Table 1 and Figure 5 of the revised manuscript, which we encourage you to review. The proposed SEAL method also demonstrates significant improvements with the 13B models, further substantiating its versatility and scalability to larger models.
>
> **References:**
>
> [1] DoRA: Weight-Decomposed Low-Rank Adaptation

---

### Author Response · Authors · 2024-11-23
**Global Response**

## Global Response

First and foremost, we extend our gratitude to each reviewer for their thorough examination of our work. In addition to this global response, we have provided detailed and appropriate answers to each reviewer and have made the necessary revisions to the manuscript (PDF). We would greatly appreciate it if the reviewers could also review the updated document.

In this study, our key contribution is the meaningful discovery that pre-trained large language models (LLMs) already possess sufficient information to handle long-context retrieval tasks, and their retrieval performance can be greatly enhanced with minimal adjustments.

As discussed in the introduction, LLMs face two primary challenges in handling long-context scenarios. Defining the model's context window size as $d$ and the input sequence length as $l$, these challenges are as follows:

1. When $l>d$, the model's performance quickly breaks down for inputs exceeding the context window size.
2. When $l<d$ but $l\approx d$, the retrieval performance degradation issue arise in handling long input contexts.

While the former issue has been addressed by techniques like context extension, the latter topic has remained largely unexplored.

In this work, **we are the first to propose methods specifically for improving retrieval performance in situations where $l<d$**. Our contributions can be summarized as follows:

1. Through circuit analysis, we demonstrate that LLMs inherently contain sufficient information to solve long-context tasks and show that simply scaling the strength of attention heads can significantly restore performance in long-context retrieval tasks.

2. We designed a pipeline to significantly improve retrieval performance at minimal cost by generating only around 50 synthetic data samples using format information from retrieval tasks and fine-tuning using those data.

3. By comparing SEAL-L, SEAL-H, and SEAL-C, we empirically validated that we accurately identified the core components of retrieval and devised the optimal method to update them, ultimately proposing a cost-effective fine-tuning technique.

As this is the first work in this research direction, there are no existing studies or appropriate baselines for comparison. We hope this point is considered. Our primary focus, as mentioned earlier, is not on proposing a more efficient PEFT method but rather on **introducing the SEAL pipeline, analyzing its impact on retrieval performance, and identifying key components through this process.**
Our SEAL pipeline is a solution that is not bound to any specific PEFT method. We hope it is recognized that SEAL-H, C, L (LoRA), and D (DoRA) are all approaches initially proposed by us.

**We acknowledge that our contributions might not have been clearly conveyed,** which may have led to the misconception that this paper is about proposing an efficient PEFT technique. To address this, we have revised the manuscript's flow and tone to improve the overall writing quality. We hope this is taken into account.

Through this research, we aim to not only significantly enhance the retrieval performance of existing models with minimal effort but also, as demonstrated in Section 7, to enable models to maintain superior performance on even longer sequences by combining our approach with context extension techniques. We believe this study contributes toward realizing such advancements effectively.

---

### Author Response · Authors · 2024-12-03
**Reminder about the end of the comments period**

First and foremost, we sincerely thank each reviewer for their effort in reviewing our work.

We would like to kindly remind you that the deadline for reviewers to leave comments is the end of December 2nd (AoE), which is approximately five hours from now.

We have provided appropriate responses to each reviewer's most recent feedback, and we hope they will be taken into account. If you have any additional feedback or concerns after reviewing our responses, we would greatly appreciate it if you could leave a comment before the deadline.

We will also do our best to respond promptly by the authors' message deadline.

---

### Meta-Review · Area_Chair_Arsg · 2024-12-20

**Metareview:**

This paper proposes an efficient method, SEAL, which tunes the scales of attention heads and channels for effective long context retrieval. Due to the limitations in the pretraining phase, LLMs are known for deficiencies in handling very long context window. By learning to weigh the strengths of different attention heads and channels using a small set of generated examples, the scaled model is capable of emphasizing the model components that are beneficial for the retrieval task and de-emphasizing those components negatively affecting the retrieval task. Experimental results demonstrate the advantage of SEAL on line retrieval task and LongBench QA task.

Strengths:
- The proposed scaling method is novel and effective, backed by an interesting observation on a significant performance change when pruning different attention heads or channels.
- Given the tunable parameters only involve scaler weights for attention heads and channels, the method is efficient and could be easily adopted in different models and real applications.
- Experimental results on two different datasets demonstrate the advantage of SEAL over the baseline.

Weaknesses:
- The clarity of the paper writing needs to be improved, such as the description of SEAL-L and SEAL-D, how the method is used in the context extension setting.
- The implication of SEAL is not clearly revealed from the empirical results. Comparing with normal PEFT methods such as LoRA and DoRA, the performance improvement is not very consistent. If the provided LoRA (SEAL-L) and DoRA (SEAL-D) are also proposed by the authors, additional baselines with normal PEFT training should be given to clearly reveal the difference and advantage of the proposed method.
- More datasets/tasks could be incorporated to showcase the method's generalizability.

**Additional Comments On Reviewer Discussion:**

- Reviewers raised concerns about more baselines and larger models, based on which the authors performed additional experiments to address the question. Results show consistent improvements over baseline LLMs. It still remains unclear whether the baselines (SEAL-L and SEAL-D) are considered as baselines or combinations with SEAL. If it is associated with SEAL, additional experiments using normal LoRA and DoRA could be carried out to serve as other baselines.
- Reviewers raised concerns regarding the details and clarity of the method and experimental setting. The authors provided further explanations.
- Reviewers also asked for experiments on other datasets and tasks. Given the limited time, the authors performed experiments on another dataset. It is suggested that further addition of long-context task datasets could be given to strengthen the claim.

---

### Decision · Program_Chairs · 2025-01-22

Reject